Investigating self-recognition in bonobos: mirror exposure reduces looking time to self but not unfamiliar conspecifics

Shorland Gladez 1
Genty Emilie 1 2
Guéry Jean-Pascal 3
Zuberbühler Klaus klaus.zuberbuehler@unine.ch 1 4
1 Department of Comparative Cognition, Institute of Biology, University of Neuchâtel , Neuchâtel , Switzerland
2 Institute of Work and Organizational Psychology, University of Neuchâtel , Neuchâtel , Switzerland
3 La Vallée des Singes Zoological park , Romagne , France
4 School of Psychology and Neuroscience, University of St. Andrews , St Andrews , Scotland (United Kingdom)
Vonk Jennifer
Electronic publication date: 2020 Aug 28
Publication date: 2020
Volume: 8
Electronic Location ID: e9685
Received 2018 May 1; Accepted 2020 Jul 18
Copyright: ©2020 Shorland et al.
Copyright year: 2020
Copyright holder: Shorland et al.
License: This is an open access article distributed under the terms of the Creative Commons Attribution License, which permits unrestricted use, distribution, reproduction and adaptation in any medium and for any purpose provided that it is properly attributed. For attribution, the original author(s), title, publication source (PeerJ) and either DOI or URL of the article must be cited.
License URL: https://creativecommons.org/licenses/by/4.0/

Keywords: Delayed self-recognition, Mental representation, Looking time, Pan paniscus, Self-awareness, Primate cognition, Social intelligence, Intelligence, Theory of mind, Evolution of cognition

Funding: European Research Council FP7/2007-2013 ERC grant 283871 Swiss National Science Foundation 31003A_166458 CR31I3_159655 The research leading to these results has received funding from the European Research Council under the European Union’s Seventh Framework Programme (FP7/2007-2013)/ERC grant agreement n° 283871 and the Swiss National Science Foundation (Social learning in primate communication: 31003A_166458 / Coordinating joint action in apes: Testing the boundaries of the human interaction engine: CR31I3_159655). The funders had no role in study design, data collection and analysis, decision to publish, or preparation of the manuscript.

==============================
The question of whether animals have some sort of self-awareness is a topic of continued debate. A necessary precondition for self-awareness is the ability to visually discriminate the self from others, which has traditionally been investigated through mirror self-recognition experiments. Although great apes generally pass such experiments, interpretations of results have remained controversial. The aim of this study was to investigate how bonobos (Pan paniscus) respond to different types of images of themselves and others, both before and after prolonged mirror exposure. We first presented presumably mirror-naive subjects with representations of themselves in three different ways (mirror image, contingent and non-contingent video footage) as well as representations of others (video footage of known and unknown conspecifics). We found that subjects paid significantly less attention to contingent images of themselves (mirror image, video footage) than to non-contingent images of themselves and unfamiliar individuals, suggesting they perceived the non-contingent self-images as novel. We then provided subjects with three months of access to a large mirror centrally positioned in the enclosure. Following this manipulation, subjects showed significantly reduced interest in the non-contingent self-images, while interest in unknown individuals remained unchanged, suggesting that the mirror experience has led to a fuller understanding of their own self. We discuss implications of this preliminary investigation for the on-going debate on self-awareness in animals.

Introduction

A fundamental question in comparative cognition is whether, or to what degree, non-human animals have something akin to self-awareness, that is, whether they can recognise themselves as separate from others and the environment. Related to this is the question of whether animals, other than humans, have some understanding of their own mental states. It is generally accepted that self-awareness presupposes self-recognition of the body, such as its visual appearance, which is empirically easier to address than mentalistic notions of the self. Since the 1970s mirrors have been used as the main tool to investigate self-recognition in the visual domain. Evidence for self-recognition is either in the form of spontaneous, self-directed, exploratory behaviours to the mirror image (Swartz, Sarauw & Evans, 1999) or subjects targeting visual markings administered to a body part that is not visible without the aid of a mirror (the “mirror-mark” test; Gallup, 1970).

Gallup’s (1970) research on chimpanzees and macaques was pioneering, followed by studies on a range of other primate species, including humans (Amsterdam, 1972), chimpanzees (Lethmate & Dücker, 1973; Suárez & Gallup, 1981; Calhoun & Thompson, 1988; Swartz & Evans, 1991; Lin, Bard & Anderson, 1992; Povinelli et al., 1993), bonobos (Hyatt & Hopkins, 1994; Westergaard & Hyatt, 1994; Walraven, Van Elsacker & Verheyen, 1995), gorillas (Suárez & Gallup, 1981; Ledbetter & Basen, 1982; Posada & Colell, 2007), orang-utans (Lethmate & Dücker, 1973; Suárez & Gallup, 1981), gibbons and siamangs (Lethmate & Dücker, 1973; Inoue-Nakamura, 1997; Suddendorf & Collier-Baker, 2009), monkeys (Lethmate & Dücker, 1973; Inoue-Nakamura, 1997) and prosimians (Inoue-Nakamura, 1997) (see Anderson, 1984 for review in primates). Within the non-human primates, great apes generally appear to be capable of mirror self-recognition (see Swartz, Sarauw & Evans, 1999 for review), although there are also multiple reports of failure (e.g., gorillas: Shillito, Gallup & Beck, 1999). In bonobos, evidence for mirror self-recognition is in terms of mirror-guided, self-directed behaviours, e.g., subjects picking the teeth or eyes (Hyatt & Hopkins, 1994; Westergaard & Hyatt, 1994), in some instances from first exposure (Walraven, Van Elsacker & Verheyen, 1995). Monkeys (Macaca silenus, Mandrillus sphinx, Papio hamadryas, Ateles sp., Cebus apella) generally fail the ‘mirror-mark’ test (Lethmate & Dücker, 1973; Ujhelyi et al., 2000; Heschl & Fuchsbichler, 2009; Suddendorf & Collier-Baker, 2009) and we are not aware of any positive evidence for spontaneous, self-directed behaviours in front of mirrors (Inoue-Nakamura, 1997). In one study, Hauser et al. (1995) used a modified ‘mirror-mark’ test by colour-dying cotton-top tamarins’ (Saguinus oedipus) head hair and reported that individuals touched their heads more often and looked in the mirror longer than controls. However, the study was criticized because results were not based on blind video scoring (Anderson & Gallup, 1997). In a follow-up study, Hauser (2001) then failed to replicate the original findings but argued that subjects had witnessed other group members with colour-dyed hair, suggesting that this may have lowered their interest.

Mirror experiments have also been conducted with non-primate species, with positive evidence in bottlenose dolphins (Marten & Psarakos, 1994; Reiss & Marino, 2001), Asian elephants (Plotnik, De Waal Frans & Reiss, 2006) and even manta rays, the biggest brained of all fish (Ari & D’Agostino, 2016). At the same time, small-brained species, such as great tits (Kraft et al., 2017) or cichlid fish (Hotta, Komiyama & Kohda, 2018), typically fail mirror self-recognition tasks, suggesting that mirror self-recognition may be a property of large brains, regardless of phylogeny (but see Gallup & Anderson, 2018). At the same time, there are a number of (disputed) claims of mirror-self recognition in Clark’s nutcrackers (Clary & Kelly, 2016), Eurasian magpies (Prior, Schwarz & Güntürkün, 2008; but see Anderson & Gallup, 2015) and cleaner wrasse (Kohda et al., 2019; but see Vonk, 2020; De Waal, 2019) but not in giant pandas (Ma et al., 2015), suggesting that the complexity of a species’ social life may also play a role (Gallup, 1998; Prior, Schwarz & Güntürkün, 2008).

In humans and great apes, the capacity to recognise one’s self in a mirror emerges gradually and with experience, usually starting with social behaviours directed at the mirror (e.g., threatening or vocalising; Gallup, 1970), followed by spatial exploration (e.g., reaching or looking behind the mirror), contingency exploration (movements of mirror-image relative to subject’s body) and self-exploration (teeth, eyes or genital regions; Swartz, Sarauw & Evans, 1999). In Western human cultures, self-directed behaviours usually appear from 15–18 months of age and become fully expressed by 24 months, while photo self-recognition occurs later (e.g., Courage, Edison & Howe, 2004; Amsterdam, 1972; Lewis & Brooks-Gunn, 1979; but see Keller et al., 2004; Keller et al., 2005; Kärtner et al., 2012 for non-western cultures). In chimpanzees, early reactions to mirrors are similar in kind but do not emerge before 24 months (Lin, Bard & Anderson, 1992). Also, there are large individual differences with either much delayed onset (around 60 months: Swartz, Sarauw & Evans, 1999) or no onset at all (e.g.,  Swartz & Evans, 1991; Povinelli et al., 1993; Walraven, Van Elsacker & Verheyen, 1995). Mirror self-recognition studies have caused much debate on how such data should be interpreted. On one end of the spectrum is the interpretation that positive evidence is an indicator of self-awareness (Inoue-Nakamura, 1997; Gallup, 1998; Swartz, Sarauw & Evans, 1999; Plotnik, De Waal Frans & Reiss, 2006) or a self-concept (“…a sense of continuity, a sense of personal agency and a sense of identity”; Gallup, 1998, p. 240). At the other end, reactions towards administered marks have been interpreted as mere artefacts of experimental manipulations, suggesting that mirror experiments reveal nothing about cognitive capacities (Heyes, 1994; Heyes, 1995; Heyes, 1996). More intermediate positions are that such behaviour qualifies as evidence for self-perception, that is, recognising one’s own visual appearance (Nielsen, Suddendorf & Slaughter, 2006) and perhaps even that one’s own body is a separate entity from the surrounding world (a ‘body concept’). The ‘body-concept’ hypothesis has been investigated in developing children, with the conclusion that such awareness emerges in the second year of life, correlated with passing the ‘mirror-mark’ test (Moore et al., 2007). The notion of a ‘body concept’ is also key to an alternative hypothesis to self-recognition: kinaesthetic-visual matching. Here, the idea is that subjects act on the contingency between the kinaesthetic sensations (caused by their body movements) and the corresponding movements of the image in the mirror (Mitchell, 1993; Mitchell, 1997). Crucially, however, mirror self-recognition does not predict perpetual kinaesthetic-visual matching; subjects may simply look at the mirror and know that what they see is them (Mitchell, 1993).

Another way of testing self-recognition is by presenting non-contingent images, usually delayed videos or simply photographs. Infants as young as 5 months appear to discriminate between video images of themselves and those of peers or objects (Legerstee, Anderson & Schaffer, 1998) and from 18–24 months they start to show the tell-tale spontaneous behaviours, such as exploration of visually inaccessible body parts, in the absence of contingency cues (Lewis & Brooks-Gunn, 1979). Between 24 and 48 months the emergence of self-recognition from non-contingent images becomes fully established, suggesting that the task is more difficult than self-recognition in a mirror (Povinelli, Landau & Perilloux, 1996). In particular, 24–36 month-old children had more difficulties to infer the presence of a sticker on their heads from watching delayed video image of themselves than 36–48 month-old children. In a more recent study, Hirata, Fuwa & Myowa (2017) used a similar variant of the ‘mark-test’ to test five subadult chimpanzees with extensive mirror experience (three of five recognised themselves in mirrors). Here, subjects were tested with live video feedback, short-delayed video feedback (i.e., 1–4 s) and long-delayed video footage of themselves (one week). Among the control conditions, the authors used video footage of humans but not of conspecifics. The finding was that the three subjects capable of mirror self-recognition removed stickers placed on their heads more effectively and exhibited more self-directed behaviours than the other two individuals, when shown live and short-delayed video feedback but not when shown long-delayed video footage or other control conditions. Although the study lacked adequate controls (Vonk, 2018), self-recognition from delayed, non-contingent self-images may be the most stringent test that an individual (human or non-human) possesses a visual mental representation of their own appearance, and this appears to require prior experience with mirrors.

In this study, we investigated whether bonobos had a generalised understanding of their own physical appearance and whether this was dependent on experience with mirrors. To address this, we tested how subjects responded to different visual images of themselves and others, including when contingency cues were absent, and how mirror exposure influenced their performance. We hypothesised that mirror-naïve individuals did not have a full understanding of their own visual appearance, but that prolonged mirror experience could provide subjects with the crucial experience. To address this, we tested mirror-naïve subjects with different visual representations of themselves and others, including non-contingent video footage of themselves. Two years later, following a 3-month period of ad libitum access to a mirror, we retested the same subjects again with the two critical conditions, i.e., non-contingent representations of themselves and unknown others. We assessed subjects’ interest in the different stimuli by comparing looking times, based on the fact that both human and nonhuman primates spend more time looking at novel than familiar faces and other stimuli (e.g., patterns: Fantz, 1964; Gunderson & Sackett, 1984; Gunderson & Swartz, 1985; objects: Bachevalier, Brickson & Hagger, 1993; Pascalis & Bachevalier, 1998; conspecific faces: Pascalis & Bachevalier, 1998; Gothard, Erickson & Amaral, 2004; Dufour, Pascalis & Petit, 2006; Gothard, Brooks & Peterson, 2009; but see Winters, Dubuc & Higham, 2015 for criticism). We predicted that stimuli perceived as unfamiliar should cause longer looking times than stimuli perceived as familiar (Pascalis & De Schonen, 1994). Mirror self-recognition experiments do not typically test whether individuals discriminate between self and others although such a design allows for direct comparisons of attention.

Method

Study site & subjects

The study was carried out at La Vallée des Singes Primate Park in Romagne (France) with a group of bonobos (2014: N = 17; 2016: N = 20; Table 1) housed in an indoor enclosure (400 m2) with access to two outdoor wooded islands (11,500 m2). Experiments were carried out from January to July 2014 and from February to July 2016. Eight subjects (Table 1) participated in all trials, which involved a ‘looking-time’ bias task with sequential stimulus presentation, a paradigm originally developed in the late 1950s for research with pre-verbal human infants (e.g., Fantz, 1963; Winters, Dubuc & Higham, 2015).

Table 1 Study subjects housed at La Vallée des Singes primate park, France.

Individual	Code	Sex	Age-Class	Year of birth	
Daniela	DNL	F	Adult	1968	
Lisala	LSL	F	Adult	1980	
Ukela	UK	F	Adult	1985	
Bondo*	BO	M	Adult	1991	
Kirembo	KI	M	Adult	1992	
Ulindi	UL	F	Adult	1993	
Diwani	DW	M	Adult	1996	
David	DV	M	Adult	2001	
Khaya	KH	F	Adult	2001	
Lingala	LNG	F	Sub-adult	2003	
Lucy*	LY	F	Sub-adult	2003	
Kelele	KEL	M	Sub-adult	2004	
Luebo*	LUE	M	Sub-adult	2006	
Nakala*	NK	F	Juvenile	2007	
Loto	LO	M	Juvenile	2009	
Moko	MO	M	Infant	2012	
Khalessi	KLS	F	Infant	2012	
Notes.

Individuals having participated in all trials are indicated in bold (N = 8). Individuals marked by an asterisk initially participated but were excluded for reasons detailed below. Age-class as defined by Kano (1984) at beginning of study.

Prior to testing, individuals were exposed to the equipment during one week to minimise any potential effects of novelty. Video stimuli were then presented by means of an APPLE iPad (screen size approx. 15 × 20 cm) placed behind a transparent Acrylic panel (Fig. 1A) to which a PANASONIC HC-V100 full HD camera was mounted to record the subjects’ reactions face-on. The mirror stimulus was presented by means of a one-way mirror (25 × 30 cm), attached behind the same panel (Figs. 1B and 1C).

Figure 1 Portable acrylic panel for stimulus presentation.

(A) Video camera and iPad displaying a non-contingent video stimulus. (B) Video camera and one-way mirror (front). (C) Video camera and one-way mirror (back). Photos by Gladez Shorland.

Part 1—before mirror exposure

The first part of this within-subject experiment was carried out in 2014 with N = 8 subjects (four males, four females, age range 9–34 years). To our knowledge, subjects had no prior experience with mirrors, but we consider it likely that some individuals already had experience with reflecting surfaces, such as window glass or bodies of water. However, reflections in mirrors are of a different quality in terms of sharpness, contrast or colour accuracy, compared to reflections in other surfaces, suggesting that subjects’ experiences with reflections of their own bodies were limited.

The experiment comprised five experimental conditions presented to all subjects in the following order: (1) contingent video footage of self (live feedback of self, visible on an iPad: C self); (2) contingent mirror image of self (live depiction of self, visible in a one-way mirror: mirror); (3) non-contingent video footage of self (previously recorded video footage of self, visible on an iPad: NC self); (4) non-contingent video footage of known group member (previously recorded video footage of other group member visible on an iPad: known); (5) non-contingent video footage of unknown conspecific (previously recorded video footage of unfamiliar conspecific visible on an iPad: unknown) (see Tables S1 and S2).

Footage for ‘NC self’ and ‘known’ were recorded during the ‘mirror’ and ‘C self’ conditions. Footage for ‘unknown’ (N = 2 males; N = 1 female) were recorded at the ‘Lola ya Bonobo’ Sanctuary in the Democratic Republic of the Congo (see Movie S1 and Movie S2 for sample footage). In both the ‘known’ and ‘unknown’ conditions, the footage showed generally inactive individuals glancing at the camera from time to time. Footage for stimulus presentation was selected at random, resulting in N = 6 same-sex pairs and N = 2 opposite-sex pairs for the ‘known’ condition and N = 4 same-sex and N = 4 opposite-sex pairs for the ‘unknown’ condition. An alternative design would have been to control for sex (i.e., to compare same-sex and different-sex footage) or to simply use same-sex footage. As this would have created sample size issues we opted for randomly assigning footage of ‘known’ and ‘unknown’ individuals to our eight subjects.

Each subject was presented with each of the five conditions once and in the same sequential order over an eight-month period (see Table S2). Stimuli were presented only very briefly, for a total of 30s, starting with the first glance from the subject towards the stimulus. A trial was terminated after 30s or as soon as a subject left. We are aware of the fact that fixed order stimulus presentation designs carry the disadvantage of potentially creating cross-condition dependencies. A completely randomised design might have been preferable but not practical due to the low number of subjects available for testing. In particular, we regarded it as essential that all subjects entered part 2 of the experiment with the exact same stimulus history. We also considered it unlikely that dependencies across conditions played a role because stimulus exposure was very short (30s) and intervals between subsequent presentations were long (median = 38.5 days, range: 0–95 days).

Trials were carried out only when a subject was alone and inactive (i.e., resting or observing) in one of the indoor cages and at a suitable orientation and distance from the corridor (0.3–2.0 m). The subject was exposed to one of two portable devices, an iPad or a one-way mirror, each with a camera mounted to record looking responses (Fig. 1). From the video clips, we extracted looking-time during stimulus presentation. Looking time as a proxy for familiarity as a whole is not uncontroversial (e.g., Winters, Dubuc & Higham, 2015). In face recognition tasks, however, looking time appears to be reliable, at least for primates, who generally look longer to unfamiliar than familiar faces of conspecifics (Pascalis & De Schonen, 1994; Pascalis & Bachevalier, 1998; Gothard, Erickson & Amaral, 2004; Gothard, Brooks & Peterson, 2009; Fujita, 1987; Demaria & Thierry, 1988). Subjects’ responses were filmed with a PANASONIC HC-V100 full HD camera, as explained before, and looking-time coded post-hoc from the video recordings. Coding was blind insofar as all videos were randomly labelled so that the rater (GS) was unable to infer the experimental condition. Videos were analysed frame-by-frame with MPEG Streamclip 1.9.2. Looking time was determined by measuring the duration between the first glance towards the stimulus and the beginning of gaze aversion. If multiple gazes occurred over the 30s stimulus presentation, we added them up. All clips were coded independently and blindly by a second rater (EG), which did not reveal any reliability issues (Pearson’s correlation coefficient, r = 0.93, N = 56). Besides looking time, we were also interested in any type of mirror-guided self-directed behaviours, especially of body parts that are visually not directly accessible, such as the face, the teeth, the eyes or the back.

Eight of 17 adult group members were tested and analysed in this study (see Table 1). Regarding the remaining individuals (N = 9), one adult male (BO) participated in all experimental conditions but had to be excluded due to poor video quality that prevented accurate coding. Two further individuals (NK and LUE) also had to be excluded because they were too close to the camera, which prevented reliable coding of looking time. One adult female (LY) had to be excluded because she participated in two conditions only. Finally, we were unable to test the remaining two adult females (DNL and UK) due to their lack of interest and participation. The dependent infants and juveniles (LO, MO and KLS) were not tested.

Part 2—after mirror exposure

Fifteen months after the first part of the experiment, the same subjects were provided with prolonged access to a large mirror (45 × 115 cm) to gain extended full experience with mirror reflections of themselves. The mirror was placed in front of a resting platform in the indoor enclosure allowing extensive ad libitum access to all individuals over a period of three months (i.e., from Nov 2015 to Feb 2016; approx. 2,000 h). At this time of year, the group was kept inside due to cold weather conditions and the park was closed to the public. Although we did not quantify the amount of time and manner by which subjects interacted with the mirror, all subjects had countless opportunities to familiarise themselves with their own mirror reflections. According to the keepers, subjects did not spend noticeable amounts of time in front of the mirror during this time, nor did they notice individuals inspecting themselves in ostensive ways. In hindsight, it would have been interesting to collect systematic data on how exactly subjects behaved in front of the mirror, both quantitatively and qualitatively.

We were mainly interested in the effect of mirror experience on subjects’ perception of their own and unknown others’ non-contingent images. To this end, we retested all individuals that had participated in part 1 (N = 8) with the two critical conditions from part 1, i.e., non-contingent video footage of themselves and of unknown conspecifics. Stimulus presentation began 10–20 days after removal of the mirror. All aspects of presentation were identical to part 1, including the video footage. The time lapse between the first and second stimulus presentation for a given subject and condition was held constant for each subject and averaged at approximately 22 months (see Table S2).

Statistical analyses

In a first analysis, prior to the 3-month mirror exposure, we modelled looking time as the response variable, experimental condition as the main predictor variable and subject ID as the random intercept in a linear mixed model (LMM). We added age as a control predictor to account for the possibility that younger individuals might show more interest than older individuals (Westergaard & Hyatt, 1994; Walraven, Van Elsacker & Verheyen, 1995). Looking time and age were square root transformed to achieve homogenous and approximately normally distributed residuals. We tested this full model against an informed null model (Forstmeier & Schielzeth, 2011), which included only age as predictor and subject ID as random intercept. We tested the difference between the full and null model with a likelihood ratio test (LRT, Dobson, 2002). Note that this comparison is equivalent to testing of ‘experimental condition’ (models with vs. without experimental condition as variable). Post hoc diagnostics were implemented in order to test the stability of the model. This entailed running the full model eight times, each time removing data of one of the eight individuals and permitted us to verify whether one of the individuals was influential with respect to the interpretation of our model results. Results of this procedure indicated stable model results, that is, exclusion of any one individual did not change the conclusions of our analysis.

In a second analysis, we tested how looking time was affected by the interaction between stimulus type (self vs. unknown, i.e., self Y/N) and mirror exposure (non-exposed vs. exposed, i.e., exposed Y/N), again with a Linear Mixed Model. Once again, we controlled for age, which was included in the respective null model, and implemented post-hoc diagnostics to test the stability of the model, which returned stable model results.

Statistical analyses were carried out with R v. 3.1.2 and lme4 v. 1.1-11 (R Core Team, 2014; Bates et al., 2015). Data are available in the electronic supplementary material (Dataset S1).

Compliance with ethical standards

All applicable international, national and institutional guidelines for the care and use of animals were followed. The study was authorised and given ethical approval by the “La Vallée des Singes” scientific coordinator and zoological director. Trials were carried out opportunistically when and where subjects felt so inclined. The study was in line with the ARRIVE guidelines and recommendations from the EAZA and the AFdPZ code of ethics.

Results

The first part of the experiment consisted of exposing subjects to different motion images of themselves, of other group members and of unfamiliar conspecifics (see Table S2). The results of the model indicated that looking time was affected by experimental condition (LRT, χ2 = 25.87, d.f. = 4, P < 0.001; Fig. 2; Table 2), with short looking times in the ‘C self’ and ‘mirror’ conditions (<5.0 of 30s) and three-fold longer looking times in the ‘NC self’ and ‘unknown’ conditions, whereas looking times to ‘known’ individuals were intermediate (Fig. 2: model estimates, Table 3: descriptive data). In other words, subjects showed most interest in images of strangers and non-contingent images of themselves and least interest in contingent images of themselves (video clips and mirror reflections), suggesting that non-contingent self-images were perceived in the same way as unknown individuals. When coding the videos for gaze duration, we did not notice any cases of self-inspecting behaviour in the two contingent conditions.

Figure 2 Subject looking time and model predictions for the five test conditions.

Subject looking time (circles) and model prediction (squares) for the five test conditions with 95% confidence interval; note that looking time was square root transformed for modeling but for presentation we back-transformed it along the y axis. Conditions are presented chronologically from left to right.

Table 2 Result of the LMM testing the effect of condition on looking time.

	Estimate	Standard error	t	
Intercept	3.18	0.44	7.18	
Condition (C self)				
−Mirror	0.01	0.49	0.02	
−NC Self	1.98	0.49	4.03	
−Known	1.03	0.49	2.08	
−Unknown	2.26	0.49	4.58	
Age	−0.12	0.03	−3.75	

Table 3 Descriptive results: experimental condition, mirror exposure, median looking-time and quartiles calculated from raw data.

Condition	Mirror exposure	Median looking time	Quartiles	
C self	non-exposed	4.6	1.4–9.3	
mirror	non-exposed	4.5	2.2–8.7	
NC self	non-exposed	22.1	11.5–26.6	
known	non-exposed	11.1	4.1–19.5	
unknown	non-exposed	18.1	14.6–27.9	
NC self	exposed	7.4	4.4–10.2	
Unknown	exposed	18.5	11.5–29.7	

We then provided subjects with three months of ad libitum mirror access. Following this manipulation, and almost two years after the previous trials, subjects were retested with non-contingent footage of themselves (NC self) and strangers (unknown), the two conditions that elicited most interest before (Fig. 2). When comparing interest before and after mirror exposure, the full model was significantly different from the null model (LRT, χ2 = 18.06, d.f. = 3, P < 0.001; Fig. 3, Table 4), indicating that looking time differed as a function of mirror-exposure, stimulus identity, and their interaction. The interaction effect was close to significance (LRT, χ2 = 3.61, df = 1, P = 0.057), as looking time substantially decreased after mirror exposure in the ‘NC self’ condition, but not in the ‘unknown’ condition (see Fig. 3 for model prediction means and SDs). Targeted follow-up tests would be necessary to confirm the preliminary conclusion that prolonged mirror exposure decreases subjects’ interest in non-contingent self-images, while interest in stranger individuals remained unchanged.

Figure 3 Subject looking time and model predictions before and after mirror exposure.

Looking time (circles) before and after three-months ad libitum mirror exposure. Model predictions (squares) are given with 95% confidence intervals; note that looking time was square root transformed for modelling but for presentation we back-transformed it along the y axis.

Table 4 Result of the LMM, testing the effects of stimulus identity (self vs. unknown) and mirror-exposure (non-exposed vs. exposed) on looking time.

	Estimate	Standard error	t	
Intercept	5.85	0.35	16.63	
Mirror-exposure (non-exposed)				
- exposed	−0.40	0.39	−1.04	
Stimulus identity (non-self)				
- self	−0.27	0.39	−0.71	
Mirror exposure: stimulus identity	−1.07	0.55	−1.96	
Age	−0.17	0.03	−6.41	

Discussion

In this study, we were interested in how prolonged mirror-exposure influenced the response to visual representations of the self. The underlying rationale was that, during human development, recognition of non-contingent footage of the self is cognitively most challenging, suggesting that bonobos may struggle with such stimuli, more than with contingent depictions of themselves. To address this, we carried out a two-part experiment during which subjects watched motion images depicting either themselves or another individual. Due to a number of constraints, discussed below, we regard the conclusions of this study as preliminary.

In the first part, subject responses to three types of self-images (mirror reflection, contingent video footage and non-contingent video footage) were compared with responses to video footage of familiar and unfamiliar conspecifics. Results revealed low interest in the mirror and in contingent self-footage, but high interest in the non-contingent self-footage condition, similar to interest in unfamiliar individuals.

A first parsimonious explanation of this result might be that bonobos did not recognise themselves in any of the three conditions (mirror, contingent, non-contingent), but that non-contingent movement was simply more interesting than contingent movement. However, this interpretation is at odds with the fact that the ‘known’ condition caused less interest than ‘non-contingent self’ and ‘unknown’ conditions, all of which moved in asynchronous ways, so we consider this explanation as unlikely.

A second explanation for the high interest in the ‘non-contingent self’ condition might be that subjects did not recognise themselves and responded as if they were unfamiliar individuals (Fig. 2) whereas they did recognise themselves in the contingent footage and mirror images. This may be because contingent self-recognition is easier to achieve than non-contingent self-recognition, to the effect that experience with low-quality reflections (such as in windows or water surfaces) could suffice to establish some level of self-recognition.

However, if subjects recognised themselves in the contingent conditions, then why did they not show the usual tell-tale behaviours of self-recognition, such as exploration of their own teeth, nose and other visually inaccessible body parts? Possible explanations for this absence is that subjects were distracted by the experimenter or by the equipment, or that stimulus exposure was simply too short (30s) to engage in self-exploration. However, it must also be stated that during the subsequent prolonged mirror exposure, neither the keepers nor the researchers noted any self-exploration by subjects, although we did not record the subjects’ behaviour in the absence of observers.

The fact that subjects showed little interest in the contingent stimuli suggests that they were already (somewhat) familiar with their own reflections, possibly from windows, water surfaces or their own shadows, but that this was not enough to develop a full sense of how one looks. Self-recognition, in other words, may not be an all-or-nothing state in bonobos, but a gradually acquired cognitive achievement.

In the second part of the experiment, we provided subjects with extended experience of self-reflections by giving them uninterrupted access to a mirror for three consecutive months. We predicted that this experience should enable subjects to familiarise themselves with a wider range of visual depictions of themselves, which would allow them to generalise and form a mental representation of their own visual appearance. We retested subjects with only the two critical stimuli (non-contingent footage of self and unknown individuals), 10–20 days after the mirror was removed, and found interest in the non-contingent footage significantly decreased, while interest in strangers remained similar (Fig. 3). In hindsight, a further interesting comparison would have been to also rerun the ‘known’ condition as a control. Based on the fact that the subjects had extensive opportunities to observe each other daily, we would not have predicted any change in interest in this condition after the mirror-exposure.

Could results be explained by low-level stimulus habituation to differences in background? We find this an unlikely scenario since stimulus presentations were exceedingly short (30s) and presented twice only over a period of 22 months, rendering perceptual habituation an implausible explanation. Related to this, subjects could have paid more attention to some backgrounds than others. Although we cannot rule this out completely, the looking time data are at odds with such an explanation: Despite similar backgrounds, responses to non-contingent self and known individuals were different, while responses to non-contingent self and strangers were similar, despite different backgrounds.

As it stands, our results are thus consistent with two hypotheses, i.e., that subjects responded to differences in perceived familiarity or that they recognized some faces as their own. At the very least, therefore, subjects must have managed to familiarise themselves with all aspects of their visual appearance, allowing them to categorise the ‘non-contingent self’ footage as more familiar compared to the stranger footage. But whether subjects really proceeded to form a true mental concept of their own visual appearance (“that’s me”), as opposed to some lower level sense of familiarity, will have to be addressed by future research. For instance, subjects could be shown (ideally contingent) footage of the stranger individuals to control for the extra time they were granted to watch themselves in the mirror. For example, subjects could be shown footage of the stranger individuals interacting with the mirror images, but such that the filmed mirror reflection only revealed their faces. If despite this extra experience they continued to show higher interest in the non-contingent footage of strangers, familiarity is the less likely explanation, suggesting that subjects understood that the individual in the video clip was them.

Conclusion

Our data suggest that, given sufficient mirror exposure, bonobos acquire the ability to use mirror-reflections of themselves to learn about their own physical appearance in more generalised way. In contrast to other research, our results suggest that this level of awareness can be detached from the here and now and can include visual non-contingent representations of the self. Whether this was achieved by responding to differences in perceived familiarity or full self-recognition cannot be conclusively decided by our investigation. Much remains to be elucidated regarding the mechanisms and the implications of great apes’ capacity of self-recognition, such as the amount of mirror exposure minimally necessary or the stability of the resulting self-recognition over time (see Calhoun & Thompson, 1988; De Veer et al., 2003).

Supplemental Information

Table S1 Experimental conditions, definition and device used for each experiment

Click here for additional data file.

Table S2 Experimental timeline - condition presentation sequence, before (part 1) and after (part 2) prolonged mirror exposure

Click here for additional data file.

Movie S1 Example footage for use as stimulus - filmed at VDS, France

Click here for additional data file.

Movie S2 Example footage for use as stimulus - filmed at Lola ya Bonobo, DRC

Click here for additional data file.

Data S1 Subject identity, age (months) and looking time (sec) for the five conditions for part 1 and two repeated conditions for part 2 of the experiment

Click here for additional data file.

We thank the keepers, Carole Michelet, Franck Alexieff, Lise Morel, Jérémy Mergault and Alexandre Albert for their valuable help in carrying out the experiment. Many thanks to Christof Neumann for his statistical advice and for valuable discussions and support.

Additional Information and Declarations

Competing Interests

Author Contributions

Animal Ethics

Data Availability

The authors declare there are no competing interests.

Gladez Shorland conceived and designed the experiments, performed the experiments, analyzed the data, prepared figures and/or tables, authored or reviewed drafts of the paper, and approved the final draft.

Emilie Genty conceived and designed the experiments, prepared figures and/or tables, authored or reviewed drafts of the paper, and approved the final draft.

Jean-Pascal Guéry performed the experiments, authored or reviewed drafts of the paper, coordinated access to study animals and gave advice on all aspects of the study, and approved the final draft.

Klaus Zuberbühler conceived and designed the experiments, authored or reviewed drafts of the paper, and approved the final draft.

The following information was supplied relating to ethical approvals (i.e., approving body and any reference numbers):

All applicable international, national, and/or institutional guidelines for the care and use of animals were followed. This experimental study was authorised and given ethical approval by the “La Vallée des Singes” scientific coordinator and zoological director.

The following information was supplied regarding data availability:

The raw dataset is available in the Supplementary File.

The video recordings of the experimental trials are provided on figshare: Shorland, Gladez; Genty, Emilie; Guéry, Jean-Pascal; Zuberbühler, Klaus (2020): Trial recordings : N=8 individuals x 7 conditions. Figshare. Media. https://doi.org/10.6084/m9.figshare.12608786.v1.

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
