# Peer review of "Investigating self-recognition in bonobos: mirror exposure reduces looking time to self but not unfamiliar conspecifics"

_PeerJ, doi:10.7717/peerj.9685_

## Round 0.1 · original submission · Major Revisions

Two expert reviewers enjoyed reading your MS and I read the paper with great interest myself. The reviewers had quite disparate views regarding the nature of your findings and the clarity with which they could speak to the issue at hand – that is whether the findings provide evidence of mirror self-recognition in bonobos. Unfortunately, I tend to agree with Reviewer 1, who suggests that your data speak only to differences in familiar versus less familiar stimuli rather than reflecting self-recognition. The entire premise for your study seems flawed because you focus on responses to novelty but allow for novelty to be confounded with self-identity. I also do not agree with the rationale for using a fixed order of stimulus presentation. You have lost the advantage of having a reasonable sample size where you could have counterbalanced the order of stimuli. In this study, there is no reason to habituate and dishabituate particular stimuli if you are looking at one time responses to different individuals in two different contexts and all of the bonobos have similar levels of prior mirror exposure. If it is possible to collect more data and include a second viewing of the familiar conspecific in comparison to the self, then I think the paper could be published. Without that added condition, I think the paper at least needs to be softened such that you are not claiming evidence of self-recognition.

I do not understand why the contingent images did not move. This removes the most important feature of the contingent video/image. Why did you display the videos for such short durations? It is possible that you can salvage your data by relating what you have found to the differences in the literature between responses to contingent and non-contingent video of the self. However, I think your data (and that of others) shows that apes are interested in moving versus non-moving stimuli. Recently Hirata and colleagues have published a paper on self-recognition in chimpanzees that involves contingent and non-contingent video taken at various delays. You should have a look at that work and reference it in your own. In my view, one of the failings in their study is that they did not present their chimpanzees with videos of familiar and/or unfamiliar conspecifics. So I appreciate that you included such conditions here. However, you would need to show that they treat familiar self video differently from familiar non-self video, which you have not done here. You do not explain why you did not re-present video of familiar individuals after exposure to the mirror.

I think you need to say more about how looking time was coded. You must report reliability for this coding.

I’d like to give you the opportunity to revise the paper; however, it will require fairly substantial revision (new data or different conclusions) before it can be considered for publication in PeerJ.

Small points:

Line 14 “investigating” should be “investigated.”
On line 84, what do you mean by “social variables”. If you mean sociality, say so.
On line 109, pluralize “position.” Line 348, pluralize “subject.”
On line 130, “fist” should be “first.”
Line 320 is missing the word “than”. Line 321 is missing “of”.

Reviewer 1 ·

Basic reporting

I enjoyed reading this paper, which is clearly written, well organized along standard lines, and provides relevant and appropriate citation and discussion of past research. Given, I assume, that the raw data file provided is not going to be published with the article, it would be helpful if the authors provided lines between the dots in Figure 2, so that the reader can see how individual bonobos responded to the different conditions. I would have liked the authors to discuss more elaborately the work with young children in western cultures showing a time lag in indicating self-recognition in response to video-self-images; what is on lines 122-127 seems ambiguous. I found Table S2 particularly helpful. I do not believe that the authors provide evidence to support their contentions about bonobo self-recognition (see below).

Experimental design

The experimental design is interesting and largely appropriate to testing differences in looking time under various conditions, and the few bonobos available easily accounts for the lack of counterbalancing in the design. However, given that the authors are suggesting that the image of the bonobo itself had become, because of the mirrors, comparable to a familiar bonobo, the design would have benefited from (1) having the bonobos also look at a video of a familiar (same sex) bonobo following the mirror intervention, and (2) collecting data on self-observation of bonobos during the mirror exposure period. And given that the authors were concerned to examine differences in responses of the bonobo to his or her own image to responses to another known or unknown bonobo, it would have made sense to have the bonobo respond to a bonobo of the same sex in all cases. In looking at the images of the unknown animals, the background is strikingly different from that of the animal itself and its known conspecifics, which introduces a further layer into the interpretation of the bonobos’ responses to the images. It would be helpful to know the dimensions of the visual image on the Apple iPad, to compare with the other two image presentations. Unfortunately, the design itself cannot provide evidence that distinguishes recognition (identification) of self, from recognition of the image of oneself as a familiar bonobo.

Validity of the findings

I do not find compelling the authors’ interpretation that the shorter post-mirror exposure looking times at the self (compared to pre-mirror exposure), but lack of difference (between pre- and post-mirror exposure) in looking times at an unknown bonobo, indicates self-recognition. I suspect that the findings could be explained by the bonobo’s recognizing the familiarity of the bonobo in the “self” image, rather than its recognizing the identity of the bonobo as itself. It seems to me that a useful test in the post-mirror phase would have included a familiar (same-sex) bonobo in the sequence shown to them. If the bonobos’ response to this image had also shown a diminishment in time spent looking, it would suggest that the “self” image was just as familiar as the known bonobo’s image. In addition, the fact that the unknown bonobo had a different background may have made it more interesting, and thus maintained the looking time of the bonobo. The fact that the bonobos did not move in the Cself and mirror conditions is frustrating, and makes comparison to the other “pre-mirror” conditions difficult to interpret, as the authors note. I would also have liked to know how much time the bonobos actually spent looking in the mirror when it was available during the 3 months; my own observations of bonobos and mirrors suggests that they don’t spend all that much time looking, but I’m sure there is variability.

Unfortunately, given that the methods cannot answer the question the authors are concerned with, namely bonobo self-recognition, I cannot see how the paper can be rehabilitated to be publishable. This is unfortunate, as it is evident that the authors put a lot of thought, time and effort into the study.

Additional comments

Specific concerns and typos:
29-30: Although true that the bonobos “may” form stable visual mental representations of themselves from interaction with a mirror over an extended period, the data do not necessarily support this interpretation: The data are consistent with the idea that bonobos may find images reflecting their own background that they see every day, and which they observed in the mirror and the NCself image, less interesting than that of an unknown bonobo in a very different background.
65-66: I suggest a more nuanced view of lesser apes’ self-recognition skills, especially as bonobos have never been tested with the mark test, so may show comparable skills (see Heschl & Fuchsbickler, 2009, who is mentioned by the authors as providing a different view, but also see Ujhelyi et al., 2000).
94: My memory is that not all chimpanzees show robustly present self-recognition (see Povinelli et al., 1993, e.g.).
95: I’d write “some individuals do not provide evidence for the capacity when tested”, otherwise it sounds like they are tested numerous times, and most aren’t.
125-127: The trend is from non-self-recognition to self-recognition here I believe.
130: “fist” should be “first”.
175: “recognize visual representations of themselves” is problematic here, as it suggests self-recognition. I suggest changing to “find visual representations of themselves familiar”.
190: What are the dimensions of the image presented on the ipad?
210-211: Given the authors’ concerns with self-recognition, it would seem same-sex images would be important.
216: Delete “upped and”.
246: “dependant” should be “dependent”.
271: “interested” should be “interest”.
285+: By using the same data in two analyses, the authors are increasing the probability of falsely detecting a difference. Given that, not taking this into account, the authors found an interaction “significant” at p = .057, I recommend they correct their alpha level appropriately, or use an alpha of .10 (which seems reasonable, given so few subjects). Effect sizes would be useful for interpretation.
303: Did age have the predicted effect? No findings for the influence of age are presented. Was age an influence at all? If not, it should be deleted as a predictor variable.
320: I don’t understand this line.
321: End line with “of”.
341: Change “indicate” to “suggest”, as there is no evidence provided that the prolonged mirror exposure is the definitive cause of any difference.
348-349: Again, I agree that the mirror MAY have allowed for a visual self-image to be developed, but I do not see that it MUST have allowed it to.
362: Change “if” to “of”.
375-377: But the authors did not test the response to a known bonobo in the post-mirror condition. In addition, in the pre-mirror condition, 6 of the 8 bonobos were in the same range for Cself and known bonobo conditions, suggesting that the suggested “intermediate” description here is not strikingly accurate.
381: Delete extra “that”.
407-413: The methods of the authors could not get rid of the simple familiarity of the bonobo’s image from the predicted effects of mirror exposure. Thus, it may be recognition of familiarity that diminished the looking time, rather than recognition of self. The authors note this on line 423, but focus on the self-recognition interpretation in the abstract and conclusion.
420, 426: The word “suggest” indicates “may” and that something “appears” to be true to the describer, so I recommend a more direct statement following “suggest”: “exposure has allowed,” “bonobos acquire”.
431: The p = .057 is not, traditionally, significant, and again, the decrease may be explained as a result of recognition of familiarity, rather than recognition of self; and the lack of decrease with the unknown bonobo may result from the interesting (and novel) background in the video.
440: Change to “bonobo’s”.
439-441: I disagree that we have gained insight into the bonobo’s “capacity to form a visual representation of self and the ability to later recognize the self-image from non-contingent video footage”; this seems an overstatement, as I don’t believe we have any new information from the research in relation to self-recognition.
445: Similarly, we already knew from earlier work that that bonobos recognize themselves.
446-447: This seems a leap.
453-455: Again, the data are mixed as to how robust self-recognition is.
458: Put comma after “species”.
460-461: Note that Mitchell (2012) suggests a different “package of cognitive capacities” related to self-recognition.
462: Change “De” to “de”.
References are not in any consistent style.
Figures 2 and 3: The legend says that the looking time was square root transformed for modeling, but the y axis states that it represents looking time in seconds, but it seems to be transformed in some way.

References mentioned:

Heschl, A., & Fuchsbickler, C. (2009). Siamangs (Hylobates syndactylus) recognize their mirror image. International Journal of Comparative Psychology, 22, 221-233.
Mitchell, R. W. (2012). Self-recognition in animals. In M. R. Leary & J. P. Tangney (Eds.), Handbook of self and identity (2nd ed., pp. 656-679). New York: Guilford.
Povinelli, D. J., Rulf, A. B., Landau, K. R., & Bierschwale, D. T. (1993). Self-recognition in chimpanzees (Pan troglodytes): Distribution, ontogeny, and patterns of emergence. Journal of Comparative Psychology, 107, 347-372.
Ujhelyi, M., Merker, B., Buk, P., & Geissmann, T. (2000). Observations on the behavior of gibbons (Hylobates leucogenys, H. Gabriellae, and H. lar) in the presence of mirrors. Journal of Comparative Psychology, 114, 253-262.

·

Basic reporting

no comments

Experimental design

I find the article very interesting, the experimental design is appropriate. I do not see any issues with using the same order of presentation; otherwise you would have introduced a factor that could have interfere with the results.

Validity of the findings

This article is interested in self recognition in non human primates. The authors have used an original paradigm based on looking time toward video instead of using the more classic and controversial mirror task. The originality of their work lies also in the fact that they tested the same participant at different time; First when they were mirror naïve then when they had been confronted to mirror for a long period.
They tested bonobos showing them contingent or non-contingent videos of themselves, videos of known or unfamiliar individuals. They were mainly interest in watching non-contingent images of themselves and of unfamiliar individuals.
After being able to see themselves in a mirror for a long period, it was found that the bonobos were less interested with their non-contingent own mirror-image compared to the video of unknown individuals.
The authors conclude that bonobos do recognize themselves in contingent videos but need training to do so in non contingent videos.
The data are clear and the authors are drawing fair conclusion.

Additional comments

I have few minor remarks:
In the introduction, there are some evidence of self recognition on moving images from 8 months of age (Legerstee et al., 1998; Child Development). I believe that this article can be cited.
Discussion: I agree with the authors that their subject might have had experience with contingency. Shadow is an other example of contingency that is naturally happening. It may explain why they are not surprised and/or interested by the contingent video. It is however evidence of self recognition and the fact that after training they are not interested anymore in the non contingent one is even showing that they understand the delay!
I think that the subject had already a representation of themselves and that the mirror experience is making this representation more familiar and stronger.
I wonder if there is any work on contingency in non human primates that can be cited here? Else, it would be interesting to discuss the fact that non contingency is an attraction parameter.

---

## Round 0.2 · Major Revisions

I appreciate that you were responsive to the comments from the last round of reviews. I have been fortunate that one of the expert reviewers from the previous round was willing to look at your revision. This reviewer provided a very thorough and thoughtful review. I think their comments are very helpful, and if you choose to take them into account, the resulting revision will greatly benefit your work. Thus, I am asking you to make another major revision to address these comments and concerns.
In addition to the reviewers' comments I have a few of my own. Line numbers refer to the tracked MS:
Although I understand you might be hesitant to cite Hauser's work, he did publish papers on MSR in tamarins including the presence of self-directed behaviors so your comment on line 66 may not be valid.
You should include estimates of error (SD or SEM) when reporting mean values (e.g., lines 331-332).
I still think you have missed the point raised by both reviewers and I in the last round about having a condition where the bonobos viewed known individuals after mirror exposure. Even if you didn't anticipate any change in their viewing behavior in this condition, it would have been nice to empirically demonstrate a lack of change in line with your predictions. That would have also further strengthened your finding of a change in the self videos.
You need to be more clear about the videos used here. You do not make it explicit that the exact same video footage for both conditions was used in both phases of the experiment. This is a very important detail. You also need to be more explicit about confounds in the stimuli as raised by the reviewers (e.g., backgrounds). The lack of control of the stimuli is a serious confound that needs to be openly addressed. Delete lines 450-452. This explanation is not relevant here.
Please use a comma after i.e. and e.g.
On line 472, please add an s to reflection.

Reviewer 1 ·

Basic reporting

Mostly fine, but see “General comments for the author.”

Experimental design

There are problems in relation to what the authors discovered and what they hoped to find. I’m also somewhat concerned as to how much the bonobos used the mirror when available for an extended time. See “General comments for the author.”

Validity of the findings

I believe that the findings are, in a limited way, valid, even with my concerns, providing the authors can say why they have such confidence that the bonobos in their study actually looked at and interacted with the mirror when available for an extended period. I also have concerns about the authors’ attempts to tie their data to so many aspects of the MSR literature, when it is really tied to a small part of that literature. See "General comments for the author.”

Additional comments

I enjoyed reading the paper, but find several problems. I believe these can be easily corrected, but it will require the authors to refocus their attention on what they actually found out, rather than making elaborate connections to the self-awareness literature (much of which I found uninformative and incorrect). The authors are arguing that mirror exposure results in knowledge of what one looks like such that one is no longer interested in looking at images of oneself. This is an interesting idea. But it means that the authors’ focus needs to be on this idea, not on issues about self-recognition per se.

Issues:

37-38: This would imply that blind people could not be self-aware.
45-79: This seems irrelevant to the research. I also found it confusing. How is the evidence for MSR in gibbons and siamangs “contradictory”? It seems some of the animals show evidence of MSR, and some don’t. Perhaps the authors meant “controversial”, as Gallup and Anderson don’t accept the evidence. I don’t see how this uncritical presentation of the literature gets us anywhere, especially because the authors are trying to provide evidence for noncontingent self-recognition, and all these studies are about contingent self-recognition.
81-91: This is a difficult topic to represent, as the evidence from American kids is most relevant to the study, but the other evidence suggests that American (and Western kids) are likely to be very different from the rest of the world. I don’t know if this literature needs to be discussed, except perhaps to indicate that photo self-recognition occurs in western children after MSR.
93-97: Again, this information would be relevant if the focus were on MSR or other forms of contingent self-recognition, but it is not the focus.
99-115: I think it is important to discuss what one gains from MSR, as this section does. I would note that Mitchell points out that MSR results in knowing what you look like, which is directly relevant to what the authors are studying. The way it is currently written, it sounds as if every time the animals recognize themselves they must use kinesthetic-visual matching; but the idea is that once they do recognize themselves, they can know “that is what I look like.”
121: I do not see this as an “alternative explanation,” but rather as an explanation for what is happening at 5 months in these infants. No one is suggesting that this is the same thing as self-recognition.
124: Not sure what “tell-tale spontaneous behavior regardless of contingency cues” is, but on 364-365, the authors write “tell-tale behaviours of self-recognition, such as exploration of visually inaccessible body parts and contingency testing”, which seems contradictory to 124.
130-132: This statement seems an overstatement. If I know what I look like, then I should recognize that the image is of me, and I might check to see if the image is an accurate depiction by seeing if I have a sticker on my head. How does this indicate that “the self possesses a temporal continuity”?
144: Delete “a sense of self”, as it adds nothing to the sentence.
161-162: Change to “themselves (i.e., without the help of kinaesthetic feedback), and”. You can’t use kinaesthetic feedback to recognize yourself in noncontingent images (or at least you can’t use kinaesthetic feedback DURING noncontingent imagery).
169-171: I agree.
186: Change “15 cm” to “15”.
207-208: An alternative would have been to simply use same-sex footage. It would have been fine to use the same male and female images for the males and females, respectively. Given that some males had male unknowns and others female unknowns, and same for females, it would be interesting to know which are which, even if you can’t do a statistical analysis.
213-214: Writing “All” here suggests that the 4 video conditions add up to 30 seconds total.
256: “we are confident”—how? Can you provide even anecdotal evidence that the animals looked in the mirror. I was stunned at how little time bonobos spent looking in the mirror when it was repeatedly provided for them in a study I worked on. So did you observe the animals looking in the mirror a lot?
271: Change to “a random intercept. We added age as a”—this adds “a” at beginning and end.
312 and elsewhere: Do you need to indicate that these numbers are means? Also, you should provide standard deviations.
324-332: I must admit that my understanding of LMM testing is limited. However, on 311-315, the author uses information from Table 2 to say that the bonobos spent more time looking at NC self and unknown than at C self and mirror. The t-tests in Table 2 supporting that these differences. However, here (324-332) the t-tests in Table 4 appear to show NO differences (except for age, about which the authors remain mute). In addition, my statistical training taught that if you use the same data in multiple analyses, you need to adjust your p-value. In the second analysis, the authors use the NC self and unknown data from the first analysis. The p-value is just over .05, but shouldn’t they be comparing it to a lower p-value? What are the effect sizes here? If large, that would be important to mention. Please make sure what I’m saying is correct by consulting a statistician before accepting it as accurate.
324, 355, and elsewhere: Change “high” to “higher” in this context. Although the animals had 30 seconds only to look, I’d say that looking for 17 seconds is not exactly “high” interest.
360-361: Exactly! Could the small size of the images have influenced this as well?
381-385: Why would water not have provided info about their physical features?
419-421: If “‘self’ is a mental construct, accessible only to its owner,” how would we ever have developed the idea? While self may be a mental construct, it is one that has a community-wide understanding (or several community-wide understandings!), and must be accessible to others to have existed as a concept in the first place.
433+: Much of this belongs in the introduction, as this is the point of the study.
448-452: But the bonobos also know the background in the NC self imagery, but not the background in the unknown. I don’t see why bringing up the difference in looking time between contingent and noncontingent conditions from the first study is relevant. There might be one explanation for one difference, and another explanation for another difference. Or perhaps I’m missing something?
455: Change to “In hindsight, it would have been desirable”.
475-477: If the evidence to support your conclusions comes from research we already know about, then why do your study? Rephrase.
491-493: Mitchell also suggests that MSR “may just be one part of a package of cognitive capacities, obtained from convergent evolution”, so not clear why “but see” is here. He presents a different suite of cognitive capacities, but a suite nonetheless, that must have occurred by convergent evolution, given dolphin and elephant MSR.
Figure 2: It would also be helpful to have the lines connecting the data points in this figure.
Figure 3: Thanks for adding the lines in Figure 3. Also, could you provide the lines connecting exposed NC self to non-exposed unknown? It is interesting to see that almost all of the bonobos lessened in their interest in the NC self image after the mirror had been present for an extended period, and that two of them did the same thing for the unknown image. The lines would allow the reader to see if there are connections among each animal’s choices.

I think that without knowing why the authors have such faith that the bonobos they tested used the mirrors over their extended presentation (and I’m willing to accept anecdotal evidence from daily observations), it’s hard to know if the extended mirror exposure had an effect. Of course it seems likely that it did. I do find the results interesting (and presume the statistical queries I posed can be easily resolved). The authors need to reframe the introduction to be about how animals would gain knowledge of what they look like from mirror exposure, such that they could be aware that the NC self image is of themselves. In that context (and I fear only in that context), the study is interesting, such that discussion of the mysterious self is really not necessary, nor is discussion of all the animals that have shown MSR and other self-related activities. The authors are presenting evidence to answer a very particular question, and their data provides little evidence beyond that answer.

---

## Round 0.3 · Minor Revisions

Thank you for your continued efforts to revise this paper. I think the introduction is much improved; clearer and more focused. However, there remain several issues with your framing of the study. While I agree that there are interesting data worth publishing here, you must be extremely careful not to misrepresent what the data can speak to given the various confounds and lack of important controls. You would do better to be completely transparent about the limitations and how they constrain interpretations of the results. If you framed the study more as a preliminary investigation that paves the way for future studies to follow up and clarify the mechanisms and repeatability of your observations, the paper would be publishable. You have done a better job of addressing concerns in the latest revision but you still gloss over the implications and embrace an interpretation that you do not have the evidence to support. Your conclusion statement is appropriate and that line of argument would serve well throughout the discussion. I do not think you need the final paragraph as your data cannot speak to this issue.
MSR experiments don’t typically test whether an individual discriminates between self and others (e.g., line 12 of the reviewing PDF). Although I do appreciate the extent to which the current study does so and think you could play up this novel aspect of your design even more.
Although long, the second paragraph of the paper is a single sentence. Paragraphs should contain opening sentences, supporting bodies and concluding statements. Combine with the next paragraph.
There are also commentaries on Kohda et al. that cast doubt on the findings in the cleaner wrasse (line 76, see de Waal, 2019 and Vonk, 2019). I have also expressed some concerns about the Hirata study discussed on lines 122-133; Vonk, 2018). Don’t feel obligated to cite these or other published critiques, but you do indicate critiques of some of the other studies earlier so it may not hurt to be consistent.
Given the importance of your subjects being mirror naïve, I think you will need to provide some indication as to why you believe that they had not been exposed to mirrors previously. What about other reflective surfaces – glass walls, windows, pools of water? If subjects were already less interested in contingent than non-contingent images of themselves before the mirror access, it suggests that they already viewed their own images as familiar, as if they had seen their own reflections before. You acknowledge this yourselves in the discussion, which undermines your own arguments concerning the strength of your mirror exposure manipulation, but you don’t attempt to reconcile this concern.
You need to make it clear that mirror exposure is not a between subjects variable and rather indicates pre/post testing, which is of course confounded with time.
On line 307, it is a bit suspect to focus on the interaction, which only approaches significance. Additionally, the interaction itself does not indicate the pattern you describe starting line 308. You would have to conduct follow-up tests to evaluate whether there are significant differences that you describe. I am sympathetic to the temptation to probe an interaction even if it is not reaching the level of statistical significance, especially given the small sample size that likely underpowered your analyses, but you at least should make it clear if those follow-up tests probing the interaction revealed significant differences in the text itself.
I think the biggest problem still is that you just do not have the adequate control condition to support your preferred interpretation of the results. Without including the familiar non-self images in the second phase, you are merely showing that they look longer at something that is less familiar, not that it matters if the more familiar faces are their own. On line 356, you gloss over the potential importance of the missing control. Perhaps an even better control would be to have shown them images of some of the unfamiliar bonobos during the mirror exposure time as well so that some images of unknown conspecifics had also become more familiar. It is also difficult to reconcile the idea that further mirror exposure made their own images more familiar when they responded to contingent video as if it was familiar in the first phase. I don’t find your attempt to reconcile these findings compelling enough.
Suddenly in the discussion, you bring up the lack of engagement in self-exploration, but you never mentioned that you recorded this or analyzed it. Indeed, you acknowledge that you did not track their behaviors during mirror exposure. Without showing that they do not engage in significant levels of that behavior, you cannot suddenly launch it as support for your hypothesis in the discussion.
Why would it be difficult to pay attention to the background but not to the stimulus itself (line 365-366)? It does not matter if the explanation is not parsimonious if it is just as feasible. What should be parsimonious is the underlying cognitive mechanism, not the explanation. You are misusing the appeal to parsimony here.
The title is still a bit misleading as we do not know that the bonobos have a perception of themselves or if they are simply responding to familiarity. Why not just “mirror exposure reduces looking time to self but not unfamiliar conspecifics” or something like that?
Small points:
Methods should be Method.
Line 164, delete the s on Videos.
Watch the placement of only throughout, e.g., line 263 should be included only…
You don’t need the “or not” on line 378.
Place commas after clauses such as after “other research” on line 377.

---

## Round 0.4 · accepted · Accept

Thank you very much for your continued work to improve the manuscript and for being so receptive to my last round of comments. I am happy to accept this version of the MS although I hope you will correct the following very minor errors during proofing:
Line 66, “witnesses” should be “witnessed.”
Line 349, change “as if it were” to “as if they were” and delete “an”
Line 374, move “only” to before “the two critical stimuli…”
Although I appreciate it very much, it is not necessary to credit the editor with suggestions made during the review process (lines 403 and 422).